Assessing an acoustic bioindicator of leisure boating disturbance on bottlenose dolphins

La Manna Gabriella glamanna@uniss.it gabriella.lamanna@gmail.com 1 2 3
Ronchetti Fabio 2
Perretti Francesco 2
Ceccherelli Giulia 1 2 3
1 National Biodiversity Future Center , Palermo , Italy
2 MareTerra - Environmental Research and Conservation , Alghero , Italy
3 Department of Chemical, Physical, Mathematical and Natural Sciences, University of Sassari , Sassari , Italy
Coscarella Mariano
Electronic publication date: 2025 Aug 1
Publication date: 2025
Volume: 13
Electronic Location ID: e19726
Received 2025 Apr 7; Accepted 2025 Jun 18
Copyright: ©2025 La Manna et al.
Copyright year: 2025
Copyright holder: La Manna et al.
License: This is an open access article distributed under the terms of the Creative Commons Attribution License, which permits unrestricted use, distribution, reproduction and adaptation in any medium and for any purpose provided that it is properly attributed. For attribution, the original author(s), title, publication source (PeerJ) and either DOI or URL of the article must be cited.
License URL: https://creativecommons.org/licenses/by/4.0/

Keywords: Acoustic monitoring, Boat traffic, Marine mammals

Funding: National Biodiversity Future Center CN00000033 Gabriella La Manna and Giulia Ceccherelli received support from the National Biodiversity Future Center funded by the Italian Ministry of University and Research, PNRR, Missione 4, Componente 2, “Dalla ricerca all’impresa”, Investimento 1.4 Project CN00000033. The funders had no role in study design, data collection and analysis, decision to publish, or preparation of the manuscript.

==============================
Growing concerns about the impact of leisure boating on marine ecosystems are particularly relevant for marine mammal species, given their dependence on sound and significant spatial overlap with human activities. Monitoring the effects of leisure boating disturbances on marine ecosystems presents considerable challenges, as it is resource-intensive and may require many years of data collection. However, since species can alter their behavior and daily routines in response to human disturbances, behaviors essential for individual fitness, reproduction, and parental care success—such as acoustic behavior—could serve as bioindicators for assessing the impact of leisure boating. Common bottlenose dolphins (Tursiops truncatus) emit acoustic signals, known as signature whistles (SWs), for individual identification, communication, and social bonding. This study aimed to examine how leisure boating influences SW characteristics (e.g., fundamental frequencies, frequency modulation, and duration) and whether these effects remain consistent across different social contexts (e.g., presence of calves), environmental conditions (e.g., seabed habitat type), and behavioral states in a population of bottlenose dolphins inhabiting the northwestern coast of Sardinia (Mediterranean Sea, Italy). The results demonstrate that certain SW characteristics were consistently affected by the presence of boats, regardless of ecological, behavioral, and social differences. These findings suggest that SWs have the potential to serve as reliable bioindicators for boating disturbances. Further research in diverse marine environments and with other dolphin populations is needed to validate and expand upon these findings.

Introduction

Among the most important human activities in coastal regions, tourism and the related sea-based recreational activities (such as leisure boating), have undergone considerable increase over the past years (Nunes & Cooke, 2021). In many regions, including the Mediterranean Sea, the nautical sector is one of the most important sources of income for coastal and insular economies, thus there is increasing concern about the detrimental impacts of leisure boating on the marine environment (Carreño & Lloret, 2021). These impacts have been classified as ‘high risk’ when they are likely to affect priority habitats and species that have already been affected by other human disturbances (such as chemical pollution, over-exploitation, climate change) (Carreño & Lloret, 2021). Additionally, the noise generated by ships and leisure boats has become the primary source of ocean noise globally, with small boats producing frequencies (100 Hz to 10,000 Hz) most sensitive to marine mammals (Houser & Finneran, 2006; Erbe et al., 2016). Moreover, leisure boats may produce the most adverse and cumulative impacts since coastal species potentially may be exposed to several vessels each day (Erbe et al., 2019).

Many studies have highlighted the negative effects of leisure boating noise on marine mammals, especially on those species which experience the highest spatial overlap with human activities: (i) masking of biologically relevant acoustic signals; (ii) avoidance from favourable habitats; (iii) stress responses; (v) temporary or permanent hearing loss; (iv) behavioral changes (reviewed in Erbe et al., 2019). Some of these responses may have implications for both individual and population fitness. Vocal production in marine mammals is energetically demanding (Holt et al., 2015), and changes in acoustic behavior—combined with increased energy expenditure from disturbance-related behavioral alterations, such as increased travel, faster movement, vessel avoidance, and displacement from preferred areas—may lead to elevated energetic costs (Erbe et al., 2019). Further, exposure to continuous disturbance may increase stress-related hormone production (Esch et al., 2009b; Rolland et al., 2012; Yang et al., 2021) suppressing individual growth, immune system function and reproduction (Romero & Butler, 2007).

Managing underwater noise presents a significant challenge due to its pervasive nature, even in areas less frequented by boats (Farcas, Thompson & Merchant, 2016). Additionally, assessing the impact of underwater noise on marine mammals is challenging due to the complex and variable nature of their responses to noise pollution (Farcas, Thompson & Merchant, 2016). However, effective management, especially within Marine Protected Areas, coupled with timely monitoring, could offer avenues for mitigating the impact of underwater noise on marine mammals (Carreño & Lloret, 2021).

Developing monitoring methods that are capable of effectively summarizing the impact of human disturbance on ecosystems, habitats and species is fundamental for environmental management and species protection (Pinna et al., 2023). In this perspective, environmental scientists have developed different ecological indicators, tools that can identify or quantify environmental impacts, based on the assumption that changes in the indicators reflect human-induced variations in a simple, time and cost-efficient way (Rombouts et al., 2013). Species-based ecological indicators (or bioindicators) usually refer to key species presence/absence, the variation in their population size and structure (number of individuals, their biomass, size and age) or their distribution range over time (Pinna et al., 2023). While monitoring the presence and abundance of marine mammals is resource-intensive and time-consuming, leveraging behavioral indicators offers a promising alternative: since species adjust their behavior in response to human disturbance, vital behavior for fitness and reproduction, such as acoustic behavior, can serve as bioindicator for monitoring disturbances caused by leisure boating (Park & Kim, 2021).

Regarding the acoustic repertoire, one of the most studied species is the common bottlenose dolphin (Tursiops truncatus) (Erbe et al., 2019), a dolphin living in complex fission–fusion societies (in which the formation and dissolution of groups are influenced by the age and sex of individuals, their behavioral activity, and reproductive conditions; (Shane, Wells & Wursig, 1986; Connor et al., 2000) for which acoustic communication is of the utmost importance (Connor et al., 2000; Aureli et al., 2008). The acoustic communication skills of the bottlenose dolphins are highly flexible: high variability can be found between different populations worldwide (Luís et al., 2021; La Manna et al., 2022). This plasticity may be due to their cosmopolitan distribution and their ability to inhabit a wide variety of different habitats: experiencing variable environmental conditions (in terms of temperature, water depth, seafloor habitat, anthropogenic disturbance) implies the need for their vocalizations to be adaptable (May-Collado & Wartzok, 2008; Larsen & Radford, 2018; La Manna et al., 2020; La Manna et al., 2022). At the same time, the differences in vocal communication may also depend on social factors such as behavior, group size, and the presence of calves (Heiler et al., 2016; La Manna et al., 2019; La Manna et al., 2020). Bottlenose dolphins produce whistles during social communication and group coordination (Janik & Sayigh, 2013). They have been classified as “variant” whistles when they are not unique to a particular individual and “signature whistles” (SWs) when they have individually distinctive frequency modulation and duration (Esch et al., 2009; Janik & Sayigh, 2013). These latter frequency-modulated and narrow-band signals are used to identify the emitter, for group contact and reunion, social bonding and vocal display (Janik & Sayigh, 2013). Bottlenose dolphins develop their SWs during the first year of their life: young dolphins learn their SWs through vocal production learning and by modelling whistles from their mother and other conspecifics (Janik & Sayigh, 2013). SWs remain approximately the same after the initial development, but dolphins may still be able to change their SWs in specific contexts (Sayigh & Janik, 2010; Sayigh et al., 2023; Labriola et al., 2025) thanks to vocal learning (Janik & Slater, 1998; King et al., 2013). Furthermore, signature whistles (SWs) are shaped by the environmental conditions of the areas in which they develop—such as water depth, substrate type, and habitat—due to the impact of these local features on sound transmission (La Manna et al., 2022). Compared to the highly changeable variant whistles, SWs are predictable and stable calling behavior which makes them good candidates as acoustic bioindicators of human disturbance. However, the use of SWs as an indicator depends on whether they exhibit observable changes in the presence of human disturbance, without being confounded by the effect of the environmental, social or behavioral contexts in which they are emitted. Thus, the aim of this study was to investigate how the presence of leisure boating (close to the group of dolphins under investigation) influences the SW contours (in term of fundamental frequencies, frequency modulation and duration) and whether this influence is consistent despite different social contexts (absence/presence of calves), environmental condition (seabed habitat type) and behavioral state of the recorded group, within a population of bottlenose dolphins inhabiting the north-western coast of Sardinia (Mediterranean Sea, Italy). We hypothesized that dolphins change the acoustic structure of their SWs in response to leisure boating disturbance, consistently in different ecological, behavioral, and social conditions. If this was true, then SWs could represent a valuable acoustic indicator of boating disturbance. Thus, this study explores the potential of using SWs to evaluate the impact of leisure boating in coastal areas.

Methods

Study area

This study took place on the north-west coast of Sardinia, Italy, off the main harbor of Alghero (40.5580°N, 8.3193°E) in an area extending for about 450 km2, with a maximum depth of 120 m (Fig. 1). The study area includes three coastal protected zones (the Capo Caccia–Isola Piana Marine Protected Area, the Site of Community Importance ‘Capo Caccia e Punta del Giglio’, and the Site of Community Importance ‘Entroterra e zona costiera tra Bosa, Capo Marargiu e Porto Tangone’), which were established due to the occurrence of endemic Mediterranean seagrass (Posidonia oceanica), coralligenous reefs and a variety of high priority species including the common bottlenose dolphin. Alghero is the largest harbor of the west Sardinian coast, and the local people rely mostly on sea-related tourism and small-scale fishery (La Manna et al., 2023a).

Figure 1 Study area.

Blue dots represent dolphin sightings in which SWs have been recorded. Black box indicates the border of the Marine Protected Area (MPA) while yellow and red boxes indicate the border of the Sites of Community Importance (SCIs). Dashed line box indicates the study area.

Field data collection

Dolphin acoustic recordings and surface behavioral data were collected during boat surveys in spring, summer, and autumn between 2013 and 2023, following the methods as previously described in La Manna et al. (2020). Systematic surveys were conducted from 9 am until 6 pm, on two identical 9.7 m motorboats powered by a 270 HP inboard engine, only in days with good sea conditions (Douglas Sea state <2 and Beaufort wind force <2) and visibility (>3 miles). To homogeneously cover the study area, routes were designed haphazardly with a generally perpendicular direction with respect to the coast and depth contours following La Manna et al. (2020). During each survey, the boat speed was kept between six and ten knots to ensure dolphin detection. Two experienced observers scanned the sea surface with naked eyes and the use of binoculars. When a sighting occurred, the navigation routes were interrupted. A dolphin sighting was defined as the observation of one or more dolphins within a visual range. When dolphins seemed to be in association by engaging in the same activity or traveling in the same direction, they were defined as a group (Shane, 1990). The group structure (age class of individuals) was recorded by two independent observers and confirmed by photo-identification analysis afterwards (as in La Manna et al., 2023a). Four different age classes were assigned as follows: (a) newborn (a calf not more than one month old, not longer than 1.20 m, always seen in the typical calf position alongside an adult); (b) calf (an individual no more than two-thirds the length of an adult); (c) juvenile (individual not fully grown, generally no more than two-thirds the length of an adult); (d) adult (fully grown individual, between 2.5 and 3 m long) (La Manna et al., 2020; La Manna et al., 2023a; La Manna et al., 2023b).

Surface behavioral and acoustic data were collected 20 min after the first encounter to allow dolphins to be habituated to the research boat presence. During each encounter, the research boat was kept at 20–100 m from the dolphins, with the engine off, while the observations lasted between 20 and 60 min. In this period, continuous focal group sampling was used to classify the surface behavioral states of dolphins during the acoustic recordings (as in La Manna et al., 2019; La Manna et al., 2020). Five behavioral states, mutually exclusive, were assigned to the dolphin group: (a) foraging (animals usually dispersed, frequent direction changes, dive intervals longer than 3 min, fish chases at the surface, birds often in attendance; (b) traveling (consistent directional movement of dolphins, with regular surfacing); (c) socializing (interactive events observed, such as body contacts, pouncing and hitting with tail, chases, aerial events, no directed movements, and variable dive intervals); (d) milling (no net movement, individuals surface facing different directions, dive intervals variable but short); and (e) resting (slow movement of dolphins, short and synchronous dive intervals) (La Manna et al., 2019; La Manna et al., 2020). When the members of the same group display different behavioral states, the predominant one (that performed by more than 50% of the group) was recorded. Furthermore, the coordinates of the sighting, the presence of newborns and calves, and the presence and number of boats within 500 m of the focal group were also recorded (La Manna et al., 2020). The maximum distance of 500 m (estimated visually) was chosen because it is expected to be the maximum distance at which engine noise from boats is considerably higher than the background noise (Sarà et al., 2007; La Manna et al., 2020). Thus, the presence of boats was used as a proxy for leisure boat noise disturbance.

The acoustic data were recorded by using a Sensor Technology SQ26-08 omnidirectional hydrophone (sensitivity–168.8 dB re 1V/µPa; flat frequency response from 100 Hz to 30 kHz, ±3 dB), with a bandwidth between 20 Hz and 50 kHz, lowered to a 5–10 m depth and connected to a ZOOM or a TASCAM recorder (data format 24-bit WAV, sampling rate 96 kHz). To minimize mechanical noise, acoustic recordings were collected only in good sea and wind conditions, and with the engine and other boat instruments switched off.

Acoustic data analysis

The acoustic recordings were analyzed in Raven Pro 1.6.5 (Bioacoustics Research Program, 2014 licensed to Gabriella La Manna; 1,024 point fast Fourier transform (FFT) and frame length, Hamming window, 50% overlap, Fs. = 48 kHz). The SIGnature Identification (SIGID) method was used to recognize signature whistles in the recordings (Janik et al., 2013). The SW of an individual is characterized by the same frequency modulation pattern (called contour). SWs can either be produced in single or multiple loops (Esch et al., 2009), characterized by repetition of the same elements and contour, usually separated by intervals less than 250 ms. A whistle was classified as an SW if a minimum of four stereotyped contours were present in a recorded session and 75% of them occurred within 1–10 s of at least one another (Janik et al., 2013). When an SW was identified, a distinctive individual code was assigned (SW-ID) and any whistle with the same contour was assigned to the same SW-ID. Each SW was graded by quality, based on its signal-to-noise ratio (SNR), as: score 1 (faint whistle with the entire contour not clearly visible on the spectrogram or overlapping with other sounds); score 2 (whistle clearly visible from its start to its end); score 3 (dominant whistle) (La Manna et al., 2013; La Manna et al., 2019; La Manna et al., 2020; La Manna et al., 2022). Only SWs with a score of 2 or 3 were included in the sample for further analysis. Further, SWs recorded when there was more than one group of dolphins present in the visual range of the observers were also excluded from the analysis. In this case, it was not possible to determine from which group the SWs were emitted, and therefore the right assignment to the surface behavior was not ensured (La Manna et al., 2020). Furthermore, only SWs recorded when boats were either absent or present within 500 m of the groups, with no other boats within visual range, were considered to ensure that the presence of boats could be used as a proxy for anthropogenic noise levels higher than the background noise. For each SW, the minimum, (the lower SW frequency) maximum (the upper SW frequency), start (the frequency measured at the start of the SW) and end frequencies (the frequency measured at the end of the SW), the number of inflection points (the number of changes from positive to negative or negative to positive slope in the contour) and the duration were automatically or manually measured by the visual inspection of the spectrogram under Raven Pro.

Statistical analysis

To examine the relationship between the SW structure and the predictors, Generalized Linear Mixed Models (GLMMs) were run using the ‘gllmTMB’ package (version 1.1.11; Brooks et al., 2017) in R Studio (version 2023.03.0+386). GLMMs are an extension of Generalized Linear Models that incorporate random effects by modeling the covariance structure resulting from data grouping (Zuur et al., 2009). They are particularly useful when data are not independent, such as when a variable is repeatedly measured from the same individuals, as is the case of SWs (La Manna et al., 2022). Three predictors were used as fixed terms in the models: (i) calf categorical variable with two levels: absent, present); (ii) behavior: categorical variable with three levels (feeding, social, travel); resting and milling were not included in the analysis due to the small number of SWs recorded during these behavioral states; (iii) habitat: categorical variable with two levels (6 mediterranean circalittoral coarse sediment, 8 biocenosis of Posidonia oceanica). SWs recorded over habitats 2 (Coralligenous biocenosis), 3 (mediterranean infralittoral coarse sediment) and 4 (mediterranean infralittoral sand) were excluded due to their small number. Information about the different habitat types were derived from the European Marine Observation Data Network (EMODnet) Seabed Habitats project (https://emodnet.ec.europa.eu/en/seabed-habitats). The fixed terms in the models were interacted with the factor boat (a categorical variable with two levels: boat present, boat absent) to investigate the influence of its presence on the association between each response variables (minimum, maximum, start and end frequency, number of inflection points and duration) and the explanatory variables. Due to the relatively high number of predictors compared to the sample size, a model including all predictors simultaneously (and the interactions with boat) resulted in overfitting. Thus, a series of simpler models was constructed. Each model included a single predictor and its interaction with boat presence, while SW-ID was considered as a random factor in each model. No model selection procedures were applied; all interaction terms between boat presence and the predictors were retained in the models to test specific a priori hypotheses about their potential modifying effects. The models were validated through graphical inspection of residuals, including residuals vs. fitted values plots to assess homogeneity, and residuals vs. explanatory variable plots to check for independence. To assess differences between all levels of categorical predictors, including the reference levels used in the GLMMs, we computed estimated marginal means and pairwise comparisons using the ‘ggemmeans’ function from the ‘ggeffects’ package (Lüdecke, 2018).

Results and Discussion

General results

A total of 275.9 h of recordings were collected over 65 sightings, and 906 high-quality signature whistles (SWs) from 54 different individuals were extracted and analyzed.

The presence of calves, behavior, and habitat—both independently and in interaction with boat presence—affected various SW acoustic variables (Table 1; Figs. 2 and 3). As expected, the marginal R2, which accounts only for the variance explained by fixed effects, was lower compared to the conditional R2, which includes both fixed and random effects, with SW_ID as the associated random effect.

Table 1 GLMMs outputs: estimated value, standard errors (SE), z-values, and significance level (p-value) for the fixed effects (explanatory variables) used in the models.

SW-ID was included as random term in each model. Marginal R2 accounts only for the variance explained by the fixed effects, while conditional R2 includes both fixed and random effects.

Max frequency	Estimate	SE	z-value	P-value	Min frequency	Estimate	SE	z-value	P-value	
(Intercept)	14.6972	0.5802	25.3310	<2e−16	(Intercept)	5.86790	0.25680	22.85300	<2e−16	
BOAT	1.1647	0.2976	3.9130	0.0001	BOAT	0.64370	0.14900	4.32100	0.00002	
CALF	0.2317	0.2663	0.8700	0.3840	CALF	0.04530	0.13310	0.34000	0.73400	
BOAT:CALF	−0.3453	0.4010	−0.8610	0.3890	BOAT:CALF	−0.96460	0.20050	−4.81200	0.00000	
R2 marg: 1%; R2 cond: 74%					R2 marg: 2%; R2 cond: 68%					
(Intercept)	15.58000	0.66750	23.34200	<2e−16	(Intercept)	6.5197	0.3006	21.691	<2e−16	
BOAT	−0.63060	0.49220	−1.28100	0.20006	BOAT	−0.2600	0.2509	−1.036	0.300084	
BEH:SOCIAL	−0.76950	0.39300	−1.95800	0.05024	BEH:SOCIAL	−0.6126	0.2004	−3.057	0.002236	
BEH:TRAVEL	−1.32820	0.46720	−2.84300	0.00447	BEH:TRAVEL	−0.8466	0.2377	−3.562	0.000368	
BOAT:SOCIAL	1.58790	0.54720	2.90200	0.00371	BOAT:SOCIAL	0.3223	0.2791	1.155	0.248093	
BOAT:TRAVEL	2.70670	0.61620	4.39300	0.00001	BOAT:TRAVEL	1.0042	0.3140	3.198	0.001385	
R2 marg: 1%; R2 cond: 75%					R2 marg: 1%; R2 cond: 68%					
(Intercept)	15.0733	0.5660	26.6310	<2e−16	(Intercept)	5.76370	0.25260	22.81700	0.00000	
BOAT	1.0251	0.2986	3.4340	0.0006	BOAT	0.41220	0.15190	2.71300	0.00666	
HABITAT	−0.8013	0.2703	−2.9640	0.0030	HABITAT	0.55810	0.13780	4.05100	0.00005	
BOAT:HABITAT	0.0187	0.4105	0.0460	0.9637	BOAT:HABITAT	0.42770	0.20910	−2.04500	0.04082	
R2 marg: 2%; R2 cond: 74%					R2 marg: 1%; R2 cond: 68%					
Start frequency	Estimate	SE	z-value	P-value	End frequency	Estimate	SE	z-value	P-value	
(Intercept)	7.19300	0.35240	20.41100	<2e−16	(Intercept)	10.6305	0.6303	16.865	<2e−16	
BOAT	0.73900	0.20660	3.57700	0.00035	BOAT	2.0383	0.3916	5.204	0.00000	
CALF	−0.37630	0.18450	−2.03900	0.04141	CALF	1.0446	0.3521	2.967	0.00301	
BOAT:CALF	−0.77180	0.27800	−2.77600	0.00550	BOAT:CALF	−1.1111	0.5295	−2.099	0.03586	
R2 marg: 2%; R2 cond: 68%					R2 marg: 2%; R2 cond: 64%					
(Intercept)	7.39121	0.41596	17.76900	<2e−16	(Intercept)	12.7793	0.7688	16.623	<2e−16	
BOAT	0.35660	0.34962	1.02000	0.30800	BOAT	−1.3113	0.6495	−2.019	0.04350	
BEH:SOCIAL	−0.39310	0.27947	−1.40700	0.16000	BEH:SOCIAL	−1.6541	0.5182	−3.192	0.00141	
BEH:TRAVEL	−0.34932	0.33102	−1.05500	0.29100	BEH:TRAVEL	−2.6439	0.6153	−4.297	0.00002	
BOAT:SOCIAL	−0.09264	0.38939	−0.23800	0.81200	BOAT:SOCIAL	2.9483	0.7221	4.083	0.00004	
BOAT:TRAVEL	0.40546	0.43734	0.92700	0.35400	BOAT:TRAVEL	4.0605	0.8130	4.994	0.00000	
R2 marg: 1%; R2 cond: 67%					R2 marg: 3%; R2 cond: 67%					
(Intercept)	6.80300	0.34930	19.47900	<2e−16	(Intercept)	11.5209	0.6307	18.267	<2e−16	
BOAT	0.85860	0.21000	4.08900	0.00004	BOAT	1.1546	0.3955	2.919	0.00351	
HABITAT	0.81160	0.19040	4.26300	0.00002	HABITAT	−1.1278	0.3590	−3.141	0.00168	
BOAT:HABITAT	−0.87270	0.28890	−3.02000	0.00252	BOAT:HABITAT	0.6770	0.5454	1.241	0.21448	
R2 marg: 2%; R2 cond: 68%					R2 marg: 1%; R2cond: 68%					
Duration	Estimate	SE	z-value	P-value	# of inflection points	Estimate	SE	z-value	P-value	
(Intercept)	−0.04440	0.03234	−1.37300	0.16973	(Intercept)	0.42183	0.17408	2.42300	0.01538	
BOAT	0.08888	0.03052	2.91200	0.00359	BOAT	−0.00627	0.08014	−0.07800	0.93767	
CALF	−0.14977	0.04750	−3.15300	0.00162	CALF	−0.19610	0.07505	−2.61300	0.00898	
BOAT:CALF	−0.04440	0.03234	−1.37300	0.16973	BOAT:CALF	0.20265	0.12098	1.67500	0.09393	
R2 marg: 1%; R2 cond: 90%					R2 marg: 1%; R2 cond: 73%					
(Intercept)	1.24482	0.13757	9.04900	<2e−16	(Intercept)	0.29707	0.19896	1.49300	0.13500	
BOAT	−0.03824	0.05601	−0.68300	0.49480	BOAT	0.16081	0.15163	1.06000	0.28900	
BEH:SOCIAL	0.07001	0.04297	1.62900	0.10320	BEH:SOCIAL	0.02184	0.11828	0.18500	0.85400	
BEH:TRAVEL	0.06522	0.05510	1.18400	0.23650	BEH:TRAVEL	0.09742	0.14517	0.67100	0.50200	
BOAT:SOCIAL	−0.11914	0.06429	−1.85300	0.06390	BOAT:SOCIAL	−0.09560	0.17177	−0.55600	0.57800	
BOAT:TRAVEL	−0.04058	0.07246	−0.56000	0.57550	BOAT:TRAVEL	−0.12349	0.18382	−0.67200	0.50200	
R2 marg: 1%; R2 cond: 90%					R2 marg: 1%; R2 cond: 73%					
(Intercept)	1.32466	0.13307	9.95400	<2e−16	(Intercept)	0.31472	0.17240	1.82500	0.06790	
BOAT	−0.14731	0.03884	−3.79300	0.00015	BOAT	0.11514	0.10013	1.15000	0.25020	
HABITAT	−0.07108	0.02991	−2.37600	0.01749	HABITAT	0.04165	0.07424	0.56100	0.57480	
BOAT:HABITAT	0.08218	0.04735	1.73600	0.08264	BOAT:HABITAT	−0.05509	0.11894	−0.46300	0.64320	
R2 marg: 1%; R2 cond: 90%					R2 marg: 1%; R2 cond: 73%					

Boat presence influenced SW maximum and end frequencies, regardless of calf presence, behavior, or habitat type, with few exceptions (Fig. 2). Specifically, maximum and end frequencies were consistently higher in the presence of boats across all social, behavioral, and environmental contexts except for SWs emitted during feeding, which remained unchanged regardless of boat presence. In contrast, minimum and start frequencies were higher in the presence of boats only in specific contexts, such as when calves were absent, during travel, or in habitat 6 (“mediterranean circalittoral coarse sediment”) (Fig. 2). Duration and the number of inflection points were not significantly affected by boat presence, social setting, behavior, or habitat type, with minor exceptions. For instance, longer SWs were emitted in the presence of calves or in habitat 6 (Fig. 3).

To our knowledge, this is one of the few studies investigating changes in SW acoustic variables under boat traffic disturbance across different behavioral, social, and environmental contexts. The observed patterns align with Perez-Ortega et al. (2021), who reported an increase in SW frequency range in response to tourist boats. Frequency shifts are among the most common acoustic adaptations dolphins use to cope with anthropogenic noise, as they enhance signal transmission efficiency and detectability (La Manna et al., 2020).

Figure 2 Effect of boat and social, behavioral and environmental contexts on the frequency-related acoustic variables of the SWs as predicted by the GLMMs (elaborated with the package “glmmTMB” in R).

Dots and bars indicate estimated marginal means and standard errors. Significant differences are indicated as: *(p < 0.05), **(p < 0.01), ***(p < 0.001). ns., not significant.

Figure 3 Effect of boat and social, behavioral and environmental contexts on the duration and number of inflection points of the SWs as predicted by the GLMM (elaborated with the package “glmmTMB” in R).

Dots and bars indicate estimated marginal means and standard errors. Significant differences are indicated as: *(p < 0.05), **(p < 0.01), ***(p < 0.001). ns., not significant.

Labriola et al. (2025) found that SWs emitted during interactions with gillnets or pots have significantly higher frequencies, likely due to the high arousal observed during cooperative feeding behaviors. Furthermore, Sayigh et al. (2023) reported higher-frequency SWs when emitted in the presence of calves, supporting the idea that adult dolphins can finely adjust their whistles depending on the situation. These recent findings and the variability in SWs we observed highlight the need for further research on individual dolphin acoustic responses to different environments, human activities, social and behavioral contexts.

Can SWs be used as acoustic bioindicator?

Directly assessing human disturbance on priority marine species can be difficult, but since animals can respond to ecosystem changes by modifying their behavior, behavior can also be considered an indicator of these changes (McDonald et al., 2017). In order to be selected as a bioindicator, a species abundance or behavior should meet the following criteria: (i) it should respond predictably to human disturbance, being sufficiently sensitive even across a subtle gradient of impact, and able of distinguishing “natural” changes from those caused by anthropogenic disturbance; (ii) it should be linked to ecologically significant impacts and, ideally, serve as an early warning indicator of change; (iii) it should be locally abundant and logistically easy to sample, or at least require less cost, time, and effort compared to other indicators (Siddig et al., 2016). Assessing human acoustic impact on marine life is difficult due to limited knowledge about many species sound production. However, the common bottlenose dolphin is well-studied, with global coastal distribution (Reynolds, Wells & Eide, 2000) and decades of research on its signature whistles (Janik & Sayigh, 2013) and, recently, on their variability across populations (Terranova et al., 2021; La Manna et al., 2022; Fandel, Silva & Bailey, 2024; Labriola et al., 2025). Further, recently SWs showed to be efficient in providing data useful to estimate ecological patterns, such as population density and spatial/temporal distribution (Romeu et al., 2024). In the present study, some SW acoustic variables change in the presence of leisure boats, despite the different ecological, behavioral, and social contexts in which the relative SWs were emitted, thus showing to be a promising bioindicator of boating disturbance. Particularly, independently of the contexts, max frequency and end frequency increased in presence of boats.

Acoustic communication reflects some functional or motivational state, signaled at some energetic cost, thus changes in acoustic communication due to chronic human disturbance (such as boating noise) may lead to sub-lethal physiological changes (e.g., increased stress levels; Esch, Sayigh & Wells, 2009). Those physiological and behavioral changes associated with the exposure to stressors which influences the health of individuals (e.g., energy metabolism or immune responses) (National Academies of Sciences, Engineering, and Medicine, 2017; Derous et al., 2020) may lead to consequences at the population level, acting on the individual survivorship and reproduction. Even if the precise role of acoustic communication in determining population fitness remains poorly understood (Andrè, 2010), human noise may impair cooperation between conspecifics through modification of acoustic behavior. Cooperation is of pivotal importance in dolphin fitness which depends on group coordination and bonding for feeding, defense against predators and parental care (Sørensen et al., 2023). While we could reasonably assume that the effect on the individual acoustic behavior can lead to consequences at population level, eventually affecting community, biodiversity and ecosystem processes (Fortin, Beyer & Boyce, 2005), the methods to demonstrate these cascading effects are still in their early stages and evidence currently available is still scarce (New et al., 2014; Erbe et al., 2022).

Thanks to technological advancements, monitoring the acoustic behavior of marine species has become more feasible compared to measuring other disturbance indicators, such as population size or distribution range, through visual surveys. Visual surveys are constrained by factors like daylight, weather conditions and the difficulty of reaching remote sites. On the contrary, acoustic monitoring has gained popularity in the last two decades because it can overcome some limitations of visual methodologies. Among the acoustic techniques to detect actively vocal marine mammals (Mellinger et al., 2007; Van Parijs et al., 2009), autonomous recorders offer several advantages. They are generally more cost-effective, capturing data over broader and finer temporal scales, and well-suited for deployment in remote areas that are difficult to survey using visual traditional methods. Additionally, they allow for continuous data collection without the risk of disrupting biological processes through human presence (Sousa-Lima et al., 2013), making them a more efficient alternative to the method employed in the present study. Further, recently, the processes of data acquisition, storage and analysis have been strongly developed. In particular, the use of Artificial Intelligence can enhance and accelerate the detection of biological signals acquired by acoustic surveys. Some recent examples have verified the efficiency of deep learning in identifying biological signals, even in the presence of ambient noise, with greater consistency than that obtained by human analysts, providing large benefit to the accuracy, efficiency, and costs of the acoustic monitoring (Ditria et al., 2022; Nur Korkmaz et al., 2023), reducing the time necessary for the analysis and interpretation of the acoustic data. Moreover, the assessment of dolphin health based on information encoded in whistle characteristics has recently been explored in captive settings through the application of machine learning methods (Jones et al., 2022). Therefore, SWs could be systematically collected over extended periods and broad geographical scales using autonomous acoustic recorders, with modern deep learning-based analysis enabling their automatic detection. Such an acoustic monitoring platform could offer a cost-effective strategy for assessing the impact of leisure boating disturbance, particularly within Marine Protected Areas. In fact, this acoustic bioindicator could play a pivotal role in monitoring and evaluating the effectiveness of conservation measures (Teixeira, Maron & van Rensburg, 2019), such as boating restrictions, by analyzing their impact on SW max and end frequencies variation in relation to the management efforts.

Even if the present results are promising, some limitations need to be underlined. In fact, the perception of noise as a risk by individuals can depend both on the status of the individuals (e.g., age, sex, health status, previous experiences), the ecological condition of the environment where the individual lives, and the concomitant effects of other disturbances (Pirotta et al., 2015; Bejder, Higham & Lusseau, 2022). Therefore, the changes in some SW acoustic variables in response to the presence of boats we found here should be validated in a range of marine environments and in other dolphin populations to assert that such responses can be generalized, thus investigating the effectiveness of SWs as an acoustic bioindicator.

Supplemental Information

Supplemental Information 1 Raw data

Special thanks to MARETERRA GROUP for the logistic support, to all the students for their help with the fieldwork, and to Lauren Polimeno for fine-tuning the English. This article reflects only the authors’ views and opinions, neither the European Union nor the European Commission can be considered responsible for them.

Additional Information and Declarations

Competing Interests

Author Contributions

Data Availability

The authors declare there are no competing interests.

Gabriella La Manna conceived and designed the experiments, performed the experiments, analyzed the data, prepared figures and/or tables, authored or reviewed drafts of the article, and approved the final draft.

Fabio Ronchetti performed the experiments, prepared figures and/or tables, authored or reviewed drafts of the article, and approved the final draft.

Francesco Perretti performed the experiments, prepared figures and/or tables, and approved the final draft.

Giulia Ceccherelli conceived and designed the experiments, authored or reviewed drafts of the article, and approved the final draft.

The following information was supplied regarding data availability:

The raw measurements are available in the Supplementary File.

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
