# Peer review of "Assessing an acoustic bioindicator of leisure boating disturbance on bottlenose dolphins"

_PeerJ, doi:10.7717/peerj.19726_

## Round 0.1 · original submission · Major Revisions

The paper is interesting and valuable, but the reviewers have identified several issues, particularly in the Methods section and the results derived from the analysis. I recommend that the authors pay close attention to Reviewer 1’s comments. If the proposed analysis cannot be performed, please clearly state this in the rebuttal letter. Additionally, provide a clearer explanation of how the models were selected.

Reviewer 1 ·

Basic reporting

Overall the paper is fairly well written but there are a few things that need tidying and I have some minor editorial suggestions which are listed below.

The reference section in particular needs tidying. The formatting of the references is inconsistent and there are a lot more references in the reference section than are cited in the text. I did some random checks because the reference list seemed very long and found quite a few examples.

The Tables and Figures also need some work:
-I think Table 1 is redundant as it almost entirely repeats the description in the text. I think it’s fine to only have it in the text, and perhaps include a bit more of the detail on habitats from the table into the text.
-Figure 1 – you refer to white dots, but they appear to be blue in the actual figure. The figure legend in general needs more information, e.g. what are the coloured boxes? The numbers don’t mean anything to the readers so provide more explanation.
Figure 2 – suggest “frequency-related acoustic variables” in the figure legend.
Figure 3 – I don’t feel strongly, but I’m not convinced this figure is necessary given the lack of significant results. You could just say the results aren’t shown because almost all were non-significant, or perhaps you could put this figure in supplemental for full transparency.

Editorial suggestions:
-Line 82: You state “Many studies” and then only cite two. I assume this is because they are reviewed in Erbe? If so then state “reviewed in Erbe et al…..”
-Lines 82-87: This sentence could do with being re-written, it’s long and not very clear.
-Lines 92-94: In which way is it controversial – I think this needs a bit more explicit explanation.
0Lines 103-104: Suggest removing “environmental” and changing end of the sentence to “simple, time and cost-efficient way”
-Line 142: Change “it” to “them”
-Line 153-156: Suggest changes to “We hypothesized that…..If this is found to be the case then SWs could represent………..explores the potential of using SWs to monitor the impact…..”
Line 168: Change to “including”
Line 229: Check tense, I think it should be was characterized.
Line 241: “while SWs with a score of 1 were excluded” is redundant
Line 248: Change to “used as a proxy”
Line 270: please add the version and citation for the R package.
Line 302: Should be Sayigh
Line 314: Not sure what is meant by sensible and generally a confusing sentence. Replace with stable?
Line 317: Perhaps change to “early warning indicator”?
Line 332: Suggest “variables change in the presence of”
Line 336: I don’t know what you mean by “honestly signaled”
Line 338: What physiological changes are you referring to that result from changes in acoustic communication? Do you perhaps mean sub-lethal energetic costs?
Line 358: Change “they” to “it”
Line 360: You introduce autonomous recorders here but I think you need to be explicit about the fact that this isn’t what you used in this study but recommend as an option for longer-term continuous monitoring. You sort of say this but it’s not very clear and I think you need to remind readers that this is different from what you did.
Line 361: More cost-effective than what? What are the traditional methods you are referring to?
Line 372-375: This is a confusing sentence

Experimental design

The research question is interesting and is clearly stated. The field and data processing steps are clearly laid out but I have concerns regarding the statistical analysis.

If I understand correctly, and is evident from Table 2, separate models were fit with each predictor variable interacting with boat presence. I’m wondering why all of the predictor variables were not included in one model given that these are contextual variables that you are trying to control for? It may be that including everything, including interactions terms between each variable and boat presence would result in an over-parameterised model, but it would be good to state that if that’s the case. It would also be valid to fit separate models if your variables are collinear, but I’d be surprised if that is the case with calf presence, behaviour, and habitat – but, of course, worth checking. Also, with factor variables, one level of the factor is taken as the intercept. In your behaviour factor this is taken as feeding. To be able to report a result for feeding you need to relevel the factor variable and run the model again with another level as the intercept. Did you do this? Because I don’t see results for feeding in Table 1 and I wonder how you got the results in Figure 2 (i.e. the NS in each case). Finally, I would expect to see model selection. In Line 267 you refer to best-fitting models but I don’t see any model selection described in the methods or evident in the results. I would, for example, expect to see the interaction term dropped from the model if not significant or not deemed to improve the model using selection criteria.

I would suggest thorough review of what you did with the analysis and expand on the details in the methods section to address my queries above so that it is clear what was done and why.

Validity of the findings

The results as presented are certainly interesting and I like the overall idea and proposal to use SW as a bioindicator for monitoring the potential effects of boat traffic. However, it’s difficult for me to assess the validity of the findings given my queries about the statistical analysis. If they are addressed and the results hold then I think this could be an interesting avenue to pursue as a monitoring tool.

Reviewer 2 ·

Basic reporting

no comment

Experimental design

no comment

Validity of the findings

no comment

Additional comments

The paper is interesting and well written. I just have some minor comments/doubts. I leave here my comments:

Introduction
Line 80: v) behavioural changes
Line 150: Why do you think the seabed is important? Is a manner of sound propagation? I would add some examples in the introduction
Line 194: In the text there is both US English “behavior” and British English “behaviour”. I would keep one version and be consistent for all the manuscript.

Methods
Lines 232-236: Here you said you were considering a SW when there were at least 4 stereotyped contours one after each other. How many whistles were you analysing from that series? Only one? The first one? I would specify it here.
Line 250: I would specify what you considered as inflection points.
I would modify Table 1 reporting just predictor and description. I would move the rest of the text in the methodology section.
Line 261: I would add “Resting and milling were not included in the analysis due to the small number of SWs recorded during these behavioral states” in the text in the methodology section and I would remove it from Table 1.
Lines 261-261: I would add this in the text in the methodology section and I would remove it from Table 1 “Seabed habitat type of the study area was a categorical variable with 5 classes, coded as follows: 2 = Coralligenous biocenosis; 3 = Mediterranean infralittoral coarse sediment; 4 = Mediterranean infralittoral sand; 6 = Mediterranean circalittoral coarse sediment; 8 = Biocenosis of Posidonia oceanica. Information about the different habitat types were derived from the European Marine Observation Data Network (EMODnet) Seabed Habitats project (https://emodnet.ec.europa.eu/en/seabed-habitats). Only SWs recorded over seabed habitats 6 and 8 were used in the analysis, while SWs recorded over habitats 2, 3 and 4 were excluded due to their small number.”
Line 270: I would add the version of R studio you used for the analysis.

Results and discussion
In the discussion part I would add more examples of change of whistles according to different behavior and habitat. In your discussion you focus a lot on the boat impact but I would add something more about habitat and behavior.
When in your dataset a SW_ID is repeated twice it means you recorded that whistle in 2 different sightings? Or even in the same sighting after a precise time of the recording?
Do you think that the fact you have some SW_ID repeated many times (SW-18-CML is repeated 84 times for example) and some repeated only two times, is affecting your results? What if a very high frequency SW whistle is recorded only in presence of boats and not in absence of boat. Is this affecting the results showing higher frequency whistles in presence of boats?
Line 277: How do you know were 56 individuals? How many SW ID did you get? 56 ID so you assume one ID corresponds to one individual? I think in your dataset I saw 54 SW_ID, does it mean that the same SW ID was shared by more than one individual?
Lines 320-329: I would cut a bit this part.
Lines 366-372: I would remove this as in your methodology there is no use of AI or automatic detector.

---

## Round 0.2 · accepted · Accept

The authors improved the manuscript based on reviewers' suggestions, and both reviewers agree with the changes.

Reviewer 1 ·

Basic reporting

This is a re-review - please see additional comments

Experimental design

This is a re-review - please see additional comments

Validity of the findings

This is a re-review - please see additional comments

Additional comments

I’m happy with the co-authors responses to my comments and queries, and happy that the description of the methods in particular is much clearer.

When re-reviewing I picked up some minor edits:

Line 115 – should be “a bioindicator”
Line 147 – should be “are a predictable”
Line 248 – remove “also”
Line 279 – suggest “interacted with boat presence”
Line 286 – should be “random effect”
Line 423 – should be “maximum”

Reviewer 2 ·

Basic reporting

In my opinion the authors improved the manuscript and replied to all my questions/suggestions. The reasearch is well presented and I think it is suitable for the publication.

Experimental design

No comment

Validity of the findings

No comment

Additional comments

No comment